# Dynamics of Public Spending on Health and Socio-Economic Development in the European Union: An Analysis from the Perspective of the Sustainable Development Goals

**DOI:** 10.3390/healthcare9030353

**Published:** 2021-03-20

**Authors:** Antonio Rafael Peña-Sánchez, José Ruiz-Chico, Mercedes Jiménez-García

**Affiliations:** Department of General Economy and INDESS, University of Cádiz, 11405 Jerez de la Frontera, Spain; jose.ruizchico@uca.es (J.R.-C.); mercedes.jimenezgarcia@uca.es (M.J.-G.)

**Keywords:** health policies, level of economic development, sigma convergence, Gini index, SDGs

## Abstract

In recent years, healthcare has become a fundamental pillar of the level of well-being of any society. With the aim of improving the lives of countries and societies, in 2015 the United Nations (UN) approved the 2030 Agenda for Sustainable Development. Among the Sustainable Development Goals (SDGs) set out in the Agenda are health and well-being (O3) and the reduction of inequalities (O10). The general objective of this paper is to analyse the impact that the level of socioeconomic development, as well as the evolution of inequalities, have had on public spending on health in European Union countries. The research methodology is based on the application of a regression model and statistical techniques such as sigma convergence, beta convergence and the Gini index. We can see that the levels of public spending on health per capita, the level of socio-economic development and the degree of inequality are closely related in these countries. For this reason, we suggest maintaining sustainable economic growth to reduce the economic disparities between EU countries, and also the current differences in public spending on health per capita.

## 1. Introduction

The incorporation of sustainability and inequality into the field of social sciences has recently led to a paradigm shift in contemporary research. Economic and social studies whose sole objective was the search for economic and utilitarian optimisation have expired. The incorporation of aspects such as sustainable development and the reduction of inequalities have marked the current path of social research. Nowadays, the sustainable nature of the productive activity is being considered more strongly, with a vision more focused on social conditions, the reduction of inequalities and environmental aspects. The current aim is mainly to establish a balance between productive activity, the environment in which it takes place, and human activity.

In 2015, the United Nations (UN) adopted the 2030 Agenda for Sustainable Development. This created a new path aimed at improving the lives of countries and societies. The Agenda incorporates a set of goals that include, among others, eradicating poverty, reducing inequalities, boosting health and well-being, protecting the planet and ensuring human prosperity [1,2,3]. These goals are specifically targeted to be achieved within the next 10 years. The wealth of projects set out in the 2013 Agenda have been grouped into the 17 Sustainable Development Goals (SDGs), with the overall aim of creating a better world not only for current generations but also for the future ones [2]. Thus, the ambitious SDG agenda requires significant investments in universal health coverage by all countries, as outlined in SDG3, health and well-being, which would undoubtedly reduce inequalities between and within countries, as envisaged in SDG10, reducing disparities [4,5].

Universal, public and quality healthcare, as a strategic economic and social sector with high added value and enormous positive externalities, is one of the basic aspects of the Welfare State [6,7,8,9,10]. One of the main concerns of the World Health Organisation (WHO) and the Member States of the European Union is equity in access to health services. In this respect, the public sector plays an essential role in trying, on the one hand, to ensure equality of opportunity among citizens and, on the other hand, to improve the level of well-being of society. Despite the generalisation of the concept, it is necessary to point out that guaranteeing equity does not imply access for all citizens to all health services at all times. There are substantial differences between states in terms of benefits, and there may even be differences in the range of services that are considered acceptable within a country (e.g., at a regional level). Each health system must have the capacity to establish criteria that clearly define a reasonable and financially affordable package of services to which to link the principle of equity of access.

The sustainability of the welfare state has recently become a controversial issue. This discussion does not stem exclusively from the period of the economic and financial crisis experienced since 2007, but dates back to the 1990s [9]. However, depending on its severity and time span, an economic crisis often affects public services such as health and education [11,12,13,14,15]. Indeed, public health seems to have been particularly vulnerable to budget cuts in the European Union, as has become evident since the economic crisis [16,17,18]. Through spending cuts and tax increases, austerity measures adopted by government authorities have hit the most economically disadvantaged population particularly hard, through the so-called “social risk effect”. This effect is caused by an increase in unemployment, poverty, homelessness, and other socio-economic risk factors (indirect effect), and the “health effect”, which comes from cuts to health services, reductions in health coverage, and restrictions on access to care (direct effect) [19,20]. Thus, health and social protection spending are often negatively associated with income inequality, so such efforts positively impact on reducing disparities [21], and reducing health spending could have severe consequences for health, especially for the most impoverished population [20].

Frequently, socio-economic analysis attempts to diagnose and examine public expenditure on health (hereafter PSH) per capita, and two conflicting ideas can be found. On the one hand, public spending is seen as an element that negatively affects the country’s economic development. Indeed, according to the recommendations of the European Union, it is necessary to establish its precise control, so as not to incur a greater public deficit and public debt, due to the economic effects that this can have not only on a specific country but also on the Eurozone. On the other hand, as an investment in health, the resources applied to this public activity are socially and economically beneficial to the country’s citizens, favouring the achievement of their activities once the potential illness that paralysed this function has been corrected.

Health and wellbeing are critical drivers of economic and social development, as they favour increased labour productivity, healthier ageing of the population, and lower spending on health-related social benefits [22,23]. Besides, advances in technological equipment can reduce social inequalities in health [24], developing CO_2_ emission policies to channel sustainable socio-economic development [25]. Public spending on health also facilitates preparedness for a possible pandemic, such as the one the planet is currently experiencing. This would enable sustainable health, as envisaged by implementing the UN SDGs, and investment in health capital to ensure a more efficient allocation of resources [5]. In this regard, research has attempted to relate health spending and economic growth under different human capital levels, concluding that when human capital is high, the economic impact on health spending and economic growth increases significantly [26].

With prior attention to the second aspect mentioned above, this paper aims to study whether health spending in European Union countries is conditioned by their level of socio-economic development. To this end, this general objective is broken down into three specific objectives: firstly, to investigate the relationship between PSH per capita and GDP per capita; secondly, to study the level of convergence experienced by these two variables; and thirdly, to analyse the evolution experienced by inequalities in the level of income per capita, and their influence on PSH in the context of the countries of the European Union.

This analysis will make it possible to observe whether the dynamics of cohesion in PSH have been similar in the European Union countries as a whole. In other words, it will help to see whether European inhabitants are in a better situation in this respect, or whether, on the contrary, the evolution of health spending has been more unequal in the period analysed (2010–2018). The latter could be causing certain disparities between people living in the same supranational territory, depending on their country of residence. This will allow us to deduce a pattern to distinguish whether health spending is higher in countries with a better socioeconomic level, or the other way around. In this sense, the level of socioeconomic development (measured by GDP per capita) can be established as a determining factor in health expenditure per capita in the European Union countries as a whole in the period under study.

In order to address the objectives of this study, we have structured the work as follows. The second section will describe the methodology used and the statistical sources consulted. The third reflects the results of the methodology applied to the data examined. The fourth section contains the discussion on the subject analysed, as well as the bibliographical sources used in the study. Finally, the last section will highlight the final conclusions from the research presented.

## 2. Materials and Methods

Due to the limited availability of homogeneous databases on health expenditure at the European level, we have consulted Eurostat, the official statistical source of the European Commission [27]. The aim was to address the objective of this scientific research with serious and rigorous data, which will enable an in-depth diagnosis of the issue addressed. However, it is necessary to point out that in the study of some variables we have found incomplete databases, with data only for a short period of time, such as the statistics on PSH, with which we only have complete data for the period 2014–2018.

In addition to the description of the data processed, the methodology used to meet the objectives set out will include an analysis of the evolution experienced by these variables and their possible links. We have examined this relationship between territorial economic development and the level of PSH in the countries of the European Union by means of an econometric estimation. We have also subjected the progress of territorial disparities to a specific study using indicators such as sigma convergence, beta convergence, the Gini index, and other applied indices to examine whether territorial differences have been reduced or, on the contrary, have intensified in the period under examination.

Sigma convergence (*σ*) is a measure of dispersion. It can be defined as the progress, over a given period, of the standard deviation of the logarithm of the *GPDpc* for the EU member states. Its expression is as follows:(1)$$\sigma_{t}={\left[\frac{\sum_{i=1}^{n}{\left[ln(GPDpc_{it})-ln(GPDpc_{t})\right]}^{2}}{n}\right]}^{\left(1/2\right)}$$
where “*ln(GPDpc_it_)*” is the logarithm of the *GPDpc* at constant prices in the *i*-th country in year “*t*”, “*ln*(*GPDpc_t_*)” is the logarithm of the *GPDpc* at constant prices of the EU countries as a whole, equivalent to a weighted average of the *GPDpc* of the member states, and “*n*” is the number of EU countries, in our case 28.

The absolute beta convergence hypothesis is tested by estimating the equation:(2)$$\ln\left(\frac{GPDpc_{i,t}}{GPDpc_{i,t-h}}\right)=\beta_{0}+\beta_{1}(GPDpc_{i,t-h})+u_{i,t}$$
where *GPDpc_i,t-1_* and *GPDpc_i,t_* represent the per capita income of the *i*-th country at the beginning and end of the period considered, respectively, *β*_0_ is the constant term, *β*_1_ is the beta coefficient and *u_i,t_* are random disturbances of zero mean and constant variance.

The econometric estimation of fixed-effects panel data can be represented as follows, trying to test the following relationship:*log*(*PSHpc_it_*) = *C_it_* + *αlog*(*GPDpc_it_*) + *u_it_*(3)
where *C_it_* is a constant that shows the influence of the rest of the elements that affect the obtaining of PSH per capita, α is the coefficient that establishes the relationship analysed, *log*(*PSHpc_it_*) is the logarithm of public health expenditure per capita of country *i* in year *t*, and log(*GPDpc_it_*) is the logarithm of the per capita income of country *i* in period *t*.

Where necessary, the data have been valued in real terms or at constant prices, by applying the GDP implicit deflator, which is clearly specified in the tables and graphs presented. In order to value GDP per capita properly, including the personal distribution within the EU countries, we have chosen to discount the concentration of GDP per capita using the following formula:*GDPpcn* = *GDPpc* * (1 − GI)(4)
where *GDPpcn* is GDP per capita net of the concentration of personal distribution, *GDPpc* is GDP per capita and GI is the Gini index, which shows the concentration of income per capita in each of the EU member countries.

On 31 January 2020, the United Kingdom’s exit from the European Union (popularly known as “Brexit”) took place. For this reason, the analysis has incorporated not only the average values for the 28-member European Union, including the United Kingdom, but also the same data for the 27-member European Union, excluding the United Kingdom. This facilitates the examination of the issue through analysis of the current composition of the European Union.

Since 2015, the year in which the UN approved the 2030 Agenda, social agents have been working to achieve the 17 goals known as SDGs. We have therefore decided to focus the study on the period 2010–2018, as it is interesting to analyse the situation and progress of these goals in the phase before the approval of the Agenda, and the dynamics experienced after its approval and implementation. The latest data officially published by the European Commission are for 2019, although for some variables there are only data up to 2018, such as PSH. The grouping into intervals of the two periods considered (2010 to 2014 and 2015 to 2018 or 2019) has been carried out to avoid the fixation of results in a single year. This technique makes it possible to overcome the deviation that could be generated in an anomalous figure. The first period considered includes the phase before the adoption of the SDGs, as well as the economic and financial crisis phase, and the second one includes the cycle after the adoption of the goals and the recovery of the economy.

## 3. Results

As explained above, the general objective of this paper is to establish the relationship between PSH and the explanatory variable GDP per capita or income per capita. Although current statistical and methodological advances make it possible to evaluate the level of economic development with other more complex and refined indicators, it is common in economic literature to use GDP per capita or income per capita as an indicator of the level of economic development. This variable is composed of the quotient between two elements: GDP in the numerator and population in the denominator. It should be noted that the population criterion is usually an element used for the distribution of public health funds in European countries. Sometimes, population ageing is also taken into account as a discriminating element in the distribution of resources [28]. Therefore, taking into account that PSH per capita includes the population in the denominator, it is considered appropriate to carry out a study of the progress experienced by the population in the countries of the European Union (Table 1).

According to data from the European Commission, the European population had 513,472 million inhabitants in 2019, with an average annual cumulative growth of 0.22%, a positive evolution in the period analysed. However, this growth has not been similar in all EU countries. In fact, there is a strong dichotomy between countries with growing populations and countries with declining ones. Among those seeing population growth are Malta, Luxembourg, Sweden, Ireland, Cyprus, Austria and the United Kingdom. However, countries such as Lithuania, Latvia, Bulgaria, Romania, Croatia, Greece, Portugal and Hungary are on the opposite side, experiencing sharp population reductions.

If we focus our attention on the proportion of the population in the European Union as a whole, there are also marked differences. Thus, those with the highest percentage of population include Germany (16%), France (13%), the United Kingdom (12%), Italy (11%), Spain (9%) and Poland (7%). While Malta (0.08%), Luxembourg (0.10%), Cyprus (0.17%), Estonia (0.26%), Latvia (0.4%), and Slovenia (0.4%) stand out for their low proportion. In the period under review, some countries have substantially increased their shares of the EU population, such as the United Kingdom (by 0.26%) and Germany and France (0.11%), and some countries have reduced it considerably, such as Romania (by 0.14%), Spain (0.13%) and Poland (0.11%).

In terms of population density, measured as the ratio between the population of each country and its surface area in square kilometres, there are considerable disparities. Thus, EU countries such as Malta, the Netherlands, Belgium, the United Kingdom, Germany, Luxembourg and Italy have a high-density ratio. Furthermore, of these countries, Malta, Luxembourg, Belgium, the United Kingdom, the Netherlands, Italy and Germany increase this variable, while other countries, such as Finland, Sweden, Estonia, Latvia, Lithuania, Bulgaria, Ireland, Croatia, Greece and Romania, mostly in Eastern Europe, have a low density. In addition, many of these countries show losses in population density, such as Lithuania, Greece, Bulgaria, Romania, Croatia and Latvia. It is interesting that Ireland is a country with a low population density, but with a growing population in the period analysed.

This clearly highlights two relevant aspects. Firstly, we can observe the disparate situation in the distribution of the population, with countries with a high volume of population compared to others with a low level of this variable. Secondly, we can see the evolution of the concentration experienced by the population in certain countries of the European Union, with a widening of the gap between the most densely populated (generally the countries of Western Europe) and the least densely populated ones (generally Eastern countries). This will also be reflected in the level of PSH in EU countries.

The dynamics of Gross Domestic Product, together with the evolution of the population, determine the progress experienced by GDP or per capita income (Table 2), the variable used in this work to assess the level of socio-economic development, as explained above. If we focus our attention on this aspect, it can be seen that GDP per capita has grown in the European Union countries as a whole by 1.2% per year cumulatively in the period under analysis. In this respect, strong divergences can be observed in the level of economic development of the member states of the European Union. Thus, countries such as Luxembourg, Ireland, Denmark, Sweden, the Netherlands, Austria, Finland, Germany, Belgium, France and the United Kingdom, have an income level per capita of over 30,000 euros in the period analysed. At the other end of the scale are Bulgaria, Romania, Croatia, Latvia, Poland, Hungary, Lithuania, Estonia and Slovakia, all Eastern European countries, with per capita GDP levels generally below 15,000 euros.

The economic and financial crisis of 2007 did not affect all EU Member States in the same way. Indeed, the countries most affected by this recession were Greece, Cyprus, Italy, Portugal, Spain, Croatia, Finland and Slovenia, which had negative cumulative average annual growth rates in the period 2010–2014, the year in which the economic recovery began. In the period 2015–2019, states that started with a lower GDP per capita grew at a higher rate. In fact, the countries that experienced the highest growth in the period indicated were Romania, Ireland, Lithuania, Poland, Hungary, Estonia, Bulgaria, Croatia, Cyprus, Latvia and Slovenia, practically all from the Eastern European area, and more recent members of the EU, with a growth of more than 3%. On the opposite side, with a growth of less than 1.5%, we can find Luxembourg, Sweden, United Kingdom, Belgium, Italy, Germany, Greece, France and Austria, countries from the West, and older in terms of the creation of the European Union.

In general, the countries with the highest growth were those that started out with the lowest level of per capita income in the period analysed (2010–2019), and vice versa. Therefore, we can deduce a priori that there has been a process of beta convergence in this decade in the level of economic development in the European Union states as a whole. According to our initial hypothesis, this level of income per capita will precisely condition the PSH of the countries of the European Union.

Concerning inequalities in the level of income per capita, based on the application of the sigma convergence indicator to the GDP per capita of the countries of the European Union in the period 1995–2019 (Figure 1), we observe that the member states tend to converge. In other words, they tend to be more equated, especially from 1999 onwards. There is also a slight upturn in divergence in the period 2008–2010, possibly because of the impact of the economic crisis, but from the latter year onwards their convergent progress continues in the period under review.

This can be seen from the application of beta convergence in the level of per capita income in the period under consideration (Figure 2). We can observe the existence of a negative relationship between the initial situation of the level of economic development of the countries of the European Union in 2010 and the economic growth experienced in the period 2010–2019. This clearly indicates a process of beta convergence in the level of GDP per capita in these countries.

If we focus on PSH, we can see that it has remained at around 80% of total health expenditure in the European Union as a whole. Moreover, it has evolved positively from 2014 to 2016. In 2017, its share decreased slightly, rising again slightly in 2018. Therefore, PSH serves the majority of the population in the EU-28 economy. Therefore, we will try to analyse the influence of the level of economic development on the evolution PSH in these countries.

PSH accounts for the largest share in countries such as Sweden, Germany, Luxembourg, Denmark, the Czech Republic, Croatia, the Netherlands and Slovakia, where it accounts for more than 80% of total health spending (Table 3). At the other end, we can find Cyprus, Greece, Bulgaria, Latvia, Portugal, Malta, Lithuania and Hungary, where the share of PSH is less than 70% of total health expenditure. It is interesting to note that in the period under analysis (2014–2018) there are countries that have significantly increased the public share of total health spending, such as France with an increase of 7%, Ireland with almost 3%, Hungary and Malta with just over 2%, Slovenia with 1.7%, and Greece and Sweden with 1%. In contrast, some countries decrease their share of public health financing, such as Portugal with a reduction of 4.5%, Estonia with 2%, the United Kingdom and Italy with just over 1.5% and Finland with 1%.

The territorial distribution of PSH is closely correlated with the spatial distribution of the population (Table 4). It is logical to think that expenditure so closely linked to the population and its structure should be distributed according to this variable. Therefore, we present the distribution of PSH per inhabitant in the EU countries. This distribution shows enormous disparities and intense crystallisation, as the ranking has hardly changed significantly over the period analysed. There are countries with PSH per capita of more than 2000 euros, such as Denmark, Sweden, Luxembourg, Germany, the Netherlands, France, Ireland, Austria, Belgium, Finland and the United Kingdom. Among these countries with the highest PSH per capita there are also strong disparities, with countries spending 4000 euros and others spending 2500 euros. On the other hand, some countries have a PSH per capita of less than 600 euros, such as Bulgaria, Romania, Latvia, Poland, Hungary and Lithuania, with a huge difference between them, some countries with an expenditure of 250 euros compared to others with 580 euros.

In addition, PSH per capita has grown by 2% in the EU-28 member states as a whole in the period under analysis (2014–2018). However, progress has not been similar in all cases and there are marked disparities. Indeed, there are countries with high cumulative average annual growth in the period 2014–2018, such as Romania, Latvia, Lithuania, Estonia, Poland, Croatia, Malta, the Czech Republic, Slovenia, Cyprus, Bulgaria and Hungary, countries from the East of the European Union, with a growth of over 3%. On the opposite side, with a growth of less than 1.5%, are countries such as Finland, which is the only one to decrease its expenditure by 0.64%, Italy, Luxembourg, the Netherlands, the United Kingdom, Belgium, Austria and Greece, the oldest countries in the European Union.

The presentation of health spending as a percentage of GDP makes it easier to detect the rate of growth of both variables (Table 5). In this respect, we can see that there is practically stagnation between the final period and the initial period analysed in the EU-28 as a whole. There are countries in the Union in which health spending per GDP is increasing, while in others this ratio is decreasing. France, Romania, Latvia, Germany, Estonia, Lithuania and Greece show a growth of more than 0.1%. In contrast, Ireland, Finland, the Netherlands, Italy, Portugal, Bulgaria, Slovakia, Hungary, the United Kingdom, Cyprus, Denmark, Belgium and Luxembourg experienced a decrease of more than 0.1% over the period.

In terms of PSH as a proportion of GDP, there are huge inequalities between the countries studied. Thus, countries such as Germany, France, Sweden, Denmark, the Netherlands, Belgium, the United Kingdom and Austria invest more than 7% of GDP in public health, although there are wide disparities between them, with differential values ranging from 7.6 to 9.60% respectively. On the other hand, Cyprus, Latvia, Bulgaria, Lithuania, Romania, Luxembourg, Poland, Hungary, Greece and Estonia, with spending of less than 5% of GDP, also have large differences, ranging from 2.8% to 4.9%. Therefore, we can affirm that there are strong disparities in the investment made by the member states of the European Union in PSH with respect to their GDP, as well as a great immobilisation of the countries in the established ranking.

As an indicator of inequality, the figure above shows the Gini index of disposable income within the EU-28 countries for the years 2010, 2015 and 2019 (Figure 3). We can see that the Gini Index has increased by 0.002 points in the EU-28 and has remained the same in the EU-27 as a whole. Moreover, in general, the countries in the East of the European Union have the highest values of the index, indicating a higher concentration of disposable income. Countries with a longer history of EU-28 membership have lower values in this index, reflecting how disposable income is more evenly distributed among their population. We will use this index to adjust GDP per capita, discounting the effect of internal inequalities, in order to establish the relationship between PSH per capita and the level of economic development in the EU-28 member countries, the main object of this scientific research.

Considering Sustainable Development Goal 10 (Reducing inequalities in the EU-28), we see that the distribution of disposable income has hardly changed in the period analysed (2010–2019) according to European Commission data. Thus, the poorest 10% of the EU population has 2.8% of disposable income, while the richest 10% of the population has 24% of disposable income. This situation has remained unchanged over the nine years analysed. Similarly, it can be seen that the poorest 20% of the population has only 7.8% of the income, while the richest 20% has nearly 40% at its disposal. This demonstrates the strong socio-economic inequalities that still exist among the population of the European Union.

A first approximation of the relationship between GDP and PSH is shown in Figure 4 and Figure 5. These represent the influence between these variables in per capita terms in the EU-28 member states in the period 2014–2018. In Figure 4 we have used GDP per capita, and in Figure 5 we have adjusted the level of economic development by discounting the influence of the Gini Index presented within the EU-28 member states, in order to penalise those countries with a high level of GDP per capita, but with a highly concentrated distribution, and vice versa. In both cases, we found a tendentially increasing relationship between the two variables, indicating that there seems to have been a close positive and direct relationship between PSH and the level of economic development in the period analysed.

In order to fulfil the main objective of this paper, we will try to establish the influence of the level of economic development (measured by GDP per capita) on PSH per capita. The regressions that establish this relationship are presented in Table 6. These estimates precisely reflect the results obtained when considering the level of economic development as a key element in the dynamics of PSH per capita for the EU-28 member states, based on panel data for the period 2014–2018.

The results of the fixed effects estimations based on panel data have been carried out, on the one hand, considering the level of economic development measured by GDP per capita as an explanatory factor (model 1) and, on the other hand, applying the adjustment to the level of economic development by discounting the inequality of the distribution of disposable income within the EU-28 member states (model 2). The application of the Hausman test gives a result that favours estimation using fixed effects. The regressions have been corrected for autocorrelation using a first-order autoregressive (AR(1)) scheme for the shocks, and for heteroscedasticity using the White procedure. The models are explanatory and the coefficients are strongly significant.

The regressions presented show that the level of GDP per capita has had a significant influence on the progress of PSH per capita in the EU-28 member states in the period under study. The coefficient on this regressor, in both estimations, is positive and highly significant at a 99% confidence level. This shows that the EU-28 countries with the highest levels of income are those that invest the most in public health (both measured in per capita terms).

## 4. Discussion

In the 1970s, the income level was already considered the main explanatory element in health expenditure. Indeed, Kleiman and Newhouse empirically demonstrated the close relationship in per capita terms between health expenditure and GDP. So nearly 90% of the variation in per capita health expenditure in 13 countries is explained by the variation in per capita GDP [29,30]. Numerous subsequent papers corroborate the empirical evidence of the link between aggregate income and health expenditure [31,32,33,34,35,36,37,38,39,40,41,42,43,44,45,46,47,48,49]. These findings have been generalised in studies in European Union countries, in the G7 and the Seven Emerging Markets (EM7) [50], in the 15 member countries of the Economic Community of West African States (ECOWAS) [51], and in countries as politically and socially distant as China and the United States [52]. We found similar positions in all of them. Moreover, not only is the link between the level of economic development and PSH investigated, but it has been established that financial development is an essential element of this variable. One study examines this relationship for a set of 159 countries over the period 1995–2014, using country-level data. The results show a close relationship between financial development and PSH and suggest policies to orient financial development towards health care to maximise the well-being of society [53].

Determinants of health expenditure include the level of income, the age structure of the population [54,55,56,57], the form of health financing (whether public or private), technological innovation [58,59], population density [60], the level of the environment [61], the degree of development of cities, the health coverage of the foreign population [62] and the number of doctors per person [58]. Widespread ideas in recent studies highlight the suggestions made to policy planners to maximise public health benefits and environmental quality, which are closely related to sustainable GDP growth.

Research based on European Social Survey (ESS) data aimed to analyse the characteristics that affect individuals and nations’ health. The most relevant finding is that personal factors such as age, economic satisfaction, level of education, unemployment, social networks, and occupational status are linked to men and women’s health. On the other hand, social factors such as PSH, lifestyle, socio-economic development, and social capital are related to people’s subjective health. Among the factors at a national level, socio-economic development measured by GDP per capita (log) is a variable closely related to improving health after controlling the characteristics individually. Finally, countries from the East of Europe highlight the worst conditions for health [63].

There are also strong disparities in health spending, as the results of this study have shown. In a previous study, it was shown that inequalities in PSH were evident in European Union member states in the period 1995–2010, indicating that these differences had become entrenched and that income per capita is the main determinant of health expenditure [64].

The impact of the economic crisis has also been relevant for health spending, due to its relationship with economic development. Indeed, during periods of economic crisis, PSH has increased to compensate for the reduction in private health spending [65]. Health spending behaves differently when GDP is growing than when it is decreasing [66]. These authors highlight that in the former case both public and private health spending increase, while in the latter case private health spending tends to stagnate while public health spending tends to increase, as the latter usually compensates for the possible decrease in private spending. Economic crises often exacerbate the challenges of demographic and technological change and major health threats. This puts the sustainability of health systems at risk and calls for the search for alternatives to enable their adequate financing [67,68,69]. A population served over time through sustainable health structures is a decisive factor for economic and social development.

It is also necessary to emphasise studies that highlight the bidirectional influence between the level of economic development (GDP per capita) and PSH per inhabitant. This research attempts to show how economic development affects PSH, and how investment in people’s health also boosts economic development by enabling people to remain active for longer and in better health [70,71,72,73]. Moreover, this public investment in health helps to reduce inequalities by limiting and hindering poverty and social exclusion [68]. In short, the findings validate the existence of a co-integrated relationship between economic growth and PSH, emphasising the bidirectional nature of the cause–effect of these economic aggregates.

Recently, the Sustainable Development Goals’ (SDG) methods to improve the world’s socio-economic and environmental conditions have led to the emergence of research that attempts to link health and well-being (O3) goals with the reduction of inequality (O10) when considering economic development. In some cases, this indicates an inconsistency between the health goals of the Sustainable Development Goals and economic growth [74]. In particular, the analysis of three East African countries (Uganda, Malawi, and Tanzania) concluded that GDP growth, which is an indicator of economic development, does not automatically benefit medical and health expenditures in different nations. They also pointed out that the facts have proved that foreign aid is not enough to make up the funding gap, as are the policy recommendations of the International Monetary Fund (IMF). International Monetary Fund loans are conditioned on GDP growth, and strict monetary and fiscal indicators have severely damaged the development of health sector expenditures. In other words, in these countries, if the government does not take sufficient measures to distribute wealth and investment equitably, and the government only focuses on GDP growth, then the existence of O8 (decent work and economic growth) centered on GDP may be detrimental, delaying efforts to achieve O3, to promote health and well-being. The health sector suggests that Governments should pay more attention to O17 (partnership to achieve goals) in global cooperation [75].

Lately, governments have reaffirmed their commitment to SDGs through the Astana Declaration, signed on 25 October 2018, which redefines the three functions of primary health care as service delivery, multisectoral action and citizen empowerment. Indeed, the health-related SDGs cannot be achieved only through the provision of health services, as some health issues are linked to other elements such as the environment, which inevitably requires agreements and joint efforts between local, national and international partners [76].

All of the above questions open up an interesting line of research, such as the creation of a sustainable health system, with a public health service capacity that allows the entire population to be adequately cared for. Some studies have already highlighted the situation in European Union member states [60], as well as the impact that Covid-19 has had on these countries in recent times. The Covid-19 pandemic has affected billions of people worldwide and has not only highlighted but also exacerbated the social and economic inequalities of recent decades [77]. This has caused new research to begin to penetrate the economic and social impact of the efficiency of research and development (R&D) spending in the health sector. An interesting study on the changes brought about by advances in research in a sample of 23 European and OECD countries for the period 2009–2017 concludes that such investment is transformed into improvements in quality of life, finding that happiness, national wealth and health spending are the factors with the greatest influence on the European population’s perception of health [78]. It also indicates that it is necessary to be prepared to deal with any type of pandemic, such as the one we are currently experiencing, relying on the implementation of the United Nations Sustainable Development Goals and investment in health care capital to ensure an efficient allocation of resources in the public health system [77,78]. However, any intervention that focuses on growing the economy in a sustainable and durable way must be accompanied by interventions that compensate for commitments to health, safety and the environment.

## 5. Conclusions

The aim of this study was to establish the relationship between the level of socio-economic development of the EU-28 member states and their public expenditure on health (PHE). This general objective has been broken down into three specific objectives. First, we have investigated the relationship between PSH and GDP, both in per capita terms. Then, we have studied the level of convergence experienced by GDP and PSH, both also at per capita level. Finally, we have analysed the evolution experienced by inequalities in income per capita, and their influence on the level of PSH in the context of the countries of the European Union. The following considerations can be drawn from this analysis.

PSH is an aspect with a significant influence on the population as a whole, guaranteeing equal opportunities among other issues. Its distribution undoubtedly enables social and territorial cohesion. In general, the public administrations of the countries with the highest levels of income per capita in the European Union tend to invest more in public health, which may be a reflection of the fact that health spending is closely linked to their economic development. However, the economic crisis of 2008 had a significant impact on PSH, affecting the population with the least economic resources with greater intensity, generating a problem of vulnerability that the current COVID-19 pandemic has aggravated. Researchers show that public investment in health should be a priority objective in any economy, as it has a positive effect on the population as a whole and reduces inequalities (O10), and ensures society’s well-being (O3). Countries’ effort in technological modernisation and human capital with a high level of competency enhances the value of health activity and even environmental improvement due to the lower generation of negative externalities, improving the sustainability of public health services.

The distribution of PSH among European geographical areas has been closely related to the territorial distribution of the population in the period analysed. However, there are still strong territorial differences in this variable per inhabitant, with member states with PSH per capita of more than 4000 euros, compared to countries with less than 500 euros. This situation highlights the difficulty experienced by the convergence of PSH per capita in the EU-28 member states, making a process of social and economic cohesion impossible. In the same way, strong disparities can be observed in the investment effort made by the EU-28 countries in health expenditure in relation to their GDP.

A graphical examination and an econometric analysis show that the level of economic development (measured by GDP per capita) is a determinant of PSH per capita in the EU Member States. To this end, two econometric regressions have been developed. In the first, GDP per capita has been considered as the explanatory variable, and in the second, GDP per capita has been taken as the regressor, discounting the concentration of income within the EU-28 countries, as measured by the Gini index. Logically, this has a fundamental impact on the process of territorial cohesion and on the equality of opportunities sought in the European economy.

The scientific study carried out shows that it would be advisable to reduce the range of dispersion between the EU-28 member countries in the level of PSH per inhabitant. To this end, the determinants of these territorial differences must be identified, as they represent a real obstacle to guaranteeing equal opportunities in access to this type of services and to achieving higher levels of well-being in the population, in line with the Sustainable Development Goals that are the subject of this research (O3 and O10). Furthermore, it is crucial to establish a system that adequately protects public health services provided by public administrations in the face of the health crisis and the fall in public revenue experienced in the EU over the last year.

Some key issues for the future need to be addressed, such as the following: (a) the sustainability of public investment in health requires medium- and long-term financial planning, with the resources foreseen in the years of economic prosperity; (b) the high territorial disparities in PSH must be rigorously controlled as they call into question equal opportunities policies; (c) a serious and rigorous debate should be opened on the level of sustainable PSH in the European Union, taking into account the restrictions established by the high public deficit and the high public indebtedness.

Finally, it should be emphasised that the research does not conclude with these reflections. It is essential to continue analysing the issue of the distribution of PSH, using new databases, updating existing ones, and applying different scientific methods, which will make it possible to contrast and consolidate the ideas put forward in this work.

## Figures and Tables

**Figure 1 healthcare-09-00353-f001:**
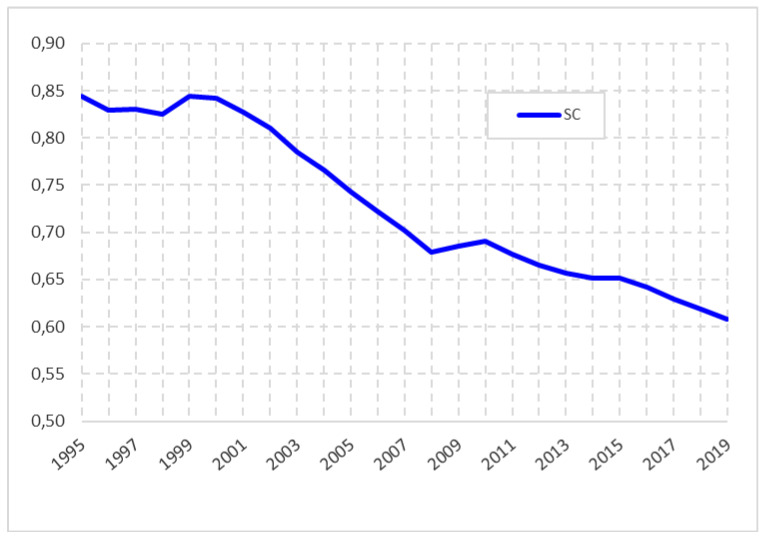
GDPpc’s Sigma convergence of EU-28 countries. Source: Prepared by authors based on Eurostat.

**Figure 2 healthcare-09-00353-f002:**
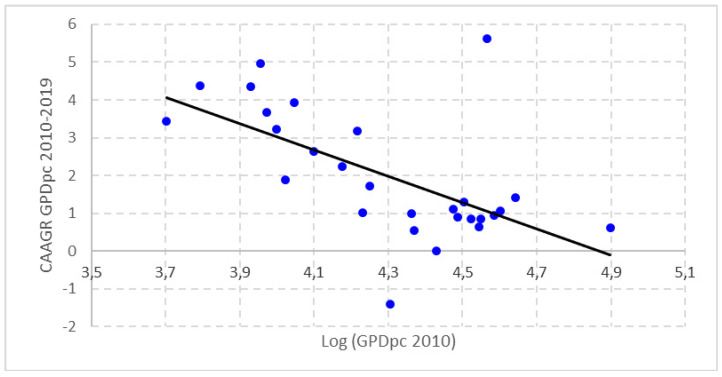
GDPpc’s Beta convergence of EU-28 countries (2010–2019). Source: Prepared by authors based on Eurostat.

**Figure 3 healthcare-09-00353-f003:**
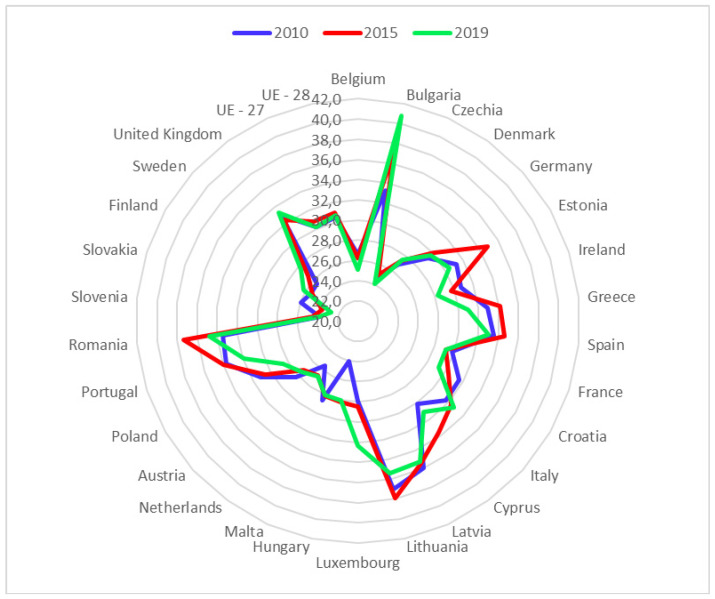
Evolution of the concentration of disposable income within the EU countries (measured by the Gini Index). Source: Prepared by authors based on Eurostat.

**Figure 4 healthcare-09-00353-f004:**
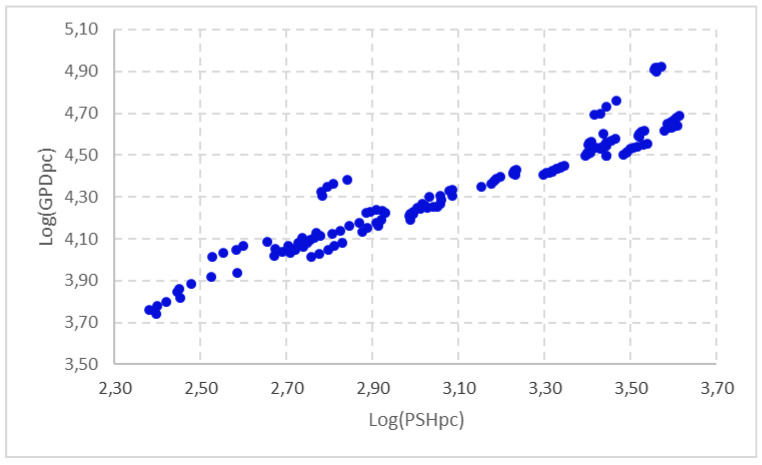
Relationship between PSH and the level of economic development of EU-28 countries (2014–2018). Source: Prepared by authors based on Eurostat.

**Figure 5 healthcare-09-00353-f005:**
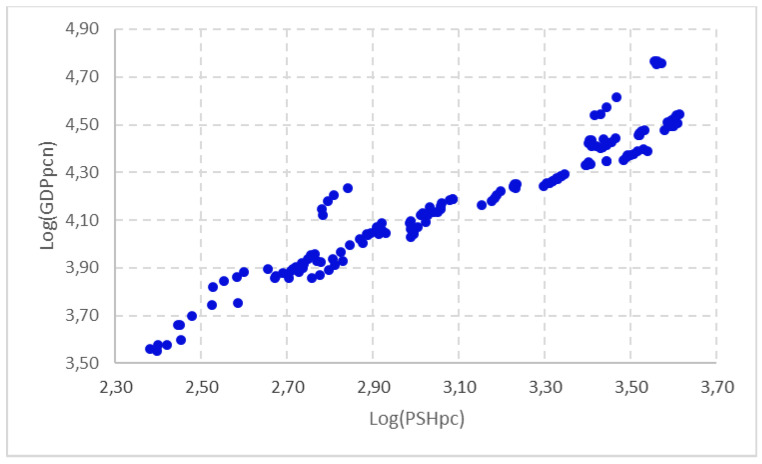
Relationship between PSH and the level of economic development adjusted with the Gini Index of EU-28 countries (2014–2018). Source: Prepared by authors based on Eurostat.

**Table 1 healthcare-09-00353-t001:** Population (number of persons), population density (number of inhabitants per square kilometre), population growth rate (%).

Countries	Population		Population Density		CAAGR ^1^	
	2010–2014	2015–2019	2010–2014	2015–2019	2010–2014	2015–2019
Belgium	11,047.049	11,350.845	361.87	368.31	0.78	0.48
Bulgaria	7329.730	7101.583	66.41	65.21	−0.60	−0.71
Czechia	10,496.562	10,586.159	133.09	133.64	0.12	0.26
Denmark	5581.149	5740.601	130.02	132.07	0.42	0.64
Germany	80,728.686	82,341.288	225.89	227.88	−0.32	0.56
Estonia	1324.832	1318.080	29.29	29.11	−0.33	0.19
Ireland	4591.445	4784.586	65.78	67.15	0.48	1.19
Greece	11,051.902	10,775.145	83.70	82.30	−0.44	−0.31
Spain	46,642.420	46,602.639	92.19	91.97	0.01	0.26
France	65,336.178	66,767.637	103.19	104.76	0.58	0.21
Croatia	4275.527	4150.387	75.55	74.49	−0.33	−0.89
Italy	59,683.387	60,578.825	197.58	200.29	0.67	−0.18
Cyprus	848.956	858.053	91.77	92.40	1.17	0.84
Latvia	2053.043	1951.903	31.79	30.76	−1.43	−0.84
Lithuania	3022.716	2852.162	46.30	44.64	−1.62	−1.11
Luxembourg	525.096	589.155	203.05	217.83	2.29	2.19
Hungary	9943.627	9806.949	106.91	105.94	−0.34	−0.21
Malta	419.699	463.933	1330.69	1396.54	0.92	2.93
Netherlands	16,714.000	17,084.920	402.36	407.17	0.38	0.56
Austria	8418.915	8747.861	100.37	102.57	0.46	0.79
Poland	38,045.954	37,979.057	121.68	121.55	0.00	−0.02
Portugal	10,520.638	10,318.674	114.07	112.64	−0.35	−0.24
Romania	20,111.425	19,644.054	84.36	83.26	−0.43	−0.58
Slovenia	2054.513	2068.149	101.34	101.74	0.17	0.22
Slovakia	5402.793	5435.297	110.18	110.57	0.12	0.13
Finland	5401.183	5498.681	15.96	16.16	0.46	0.21
Sweden	9487.973	9988.790	21.63	22.25	0.80	1.22
United Kingdom	63,456.872	65,799.453	261.68	267.50	0.73	0.68
UE-27	441,059.397	445,385.412	104.64	105.25	0.13	0.18
UE-28	504,516.304	511,184.865	113.18	114.08	0.20	0.24

^1^ Cumulative average annual growth rate. Source: Prepared by authors based on Eurostat.

**Table 2 healthcare-09-00353-t002:** Gross Domestic Product per capita (2010 constant prices in purchasing power parity).

Countries	GDPpc ^1^		CAAGR ^2^	
	2010–2014	2015–2019	2010–2014	2015–2019
Belgium	33.528	35.094	0.40	1.13
Bulgaria	5.326	6.308	2.30	4.25
Czechia	15.228	17.354	0.76	2.99
Denmark	44.310	47.668	0.59	2.17
Germany	33.134	35.136	1.52	1.23
Estonia	12.258	14.474	4.03	4.28
Ireland	37.456	54.256	2.08	5.00
Greece	17.728	17.194	−4.42	1.28
Spain	22.388	24.276	−0.91	2.22
France	31.110	32.364	0.51	1.34
Croatia	10.394	11.564	−0.50	4.03
Italy	26.188	26.370	−1.43	1.23
Cyprus	21.746	23.036	−3.55	3.98
Latvia	9.576	11.650	4.83	3.77
Lithuania	10.260	12.770	5.68	4.79
Luxembourg	78.646	82.768	0.10	0.71
Hungary	10.262	12.128	1.97	4.29
Malta	17.224	20.960	3.15	2.28
Netherlands	38.490	40.606	0.07	1.68
Austria	36.078	37.106	0.52	1.38
Poland	9.954	11.868	2.66	4.53
Portugal	16.426	17.612	−1.09	2.84
Romania	6.572	8.210	3.23	5.73
Slovenia	17.552	19.378	−0.18	3.57
Slovakia	13.138	15.036	2.06	2.68
Finland	35.016	36.024	−0.50	1.91
Sweden	40.588	43.318	0.76	0.77
United Kingdom	30.386	32.364	1.20	0.88
UE-27	25.144	26.990	0.50	1.92
UE-28	25.808	27.682	0.62	1.77

^1^ Gross Domestic Product per capita. ^2^ Cumulative average annual growth rate. Source: Prepared by authors based on Eurostat.

**Table 3 healthcare-09-00353-t003:** Public expenditure on health (PSH) in EU Member States (as % of Total Spending on Health).

Countries	2014	2015	2016	2017	2018
Belgium	75.85	75.77	76.10	75.96	75.81
Bulgaria	58.48	56.12	55.36	55.94	59.08
Czechia	82.69	82.37	81.98	82.09	83.03
Denmark	84.19	84.18	84.12	84.02	83.88
Germany	84.21	84.17	84.31	84.52	84.56
Estonia	75.68	75.62	75.66	73.59	73.67
Ireland	71.02	72.04	72.44	72.79	73.89
Greece	57.67	57.58	60.74	60.46	58.75
Spain	70.27	71.32	71.55	70.65	70.40
France	76.54	76.64	83.07	83.27	83.63
Croatia	82.92	82.63	82.52	82.54	82.84
Italy	75.42	74.44	74.40	73.74	73.89
Cyprus	43.05	41.60	41.09	41.35	42.99
Latvia	59.67	58.65	55.87	57.33	59.88
Lithuania	67.56	67.14	66.60	66.14	67.05
Luxembourg	83.48	83.66	83.61	83.95	84.08
Hungary	67.10	68.19	68.10	69.14	69.45
Malta	61.32	60.57	63.03	62.90	63.48
Netherlands	81.08	81.38	81.14	81.65	82.07
Austria	74.01	74.06	73.96	73.98	74.70
Poland	70.66	69.99	69.85	69.29	71.49
Portugal	66.08	66.17	61.48	61.11	61.54
Romania	79.00	78.02	78.33	78.65	79.73
Slovenia	71.11	71.81	72.73	72.20	72.82
Slovakia	80.23	79.72	80.36	79.94	80.13
Finland	78.02	76.94	76.23	76.42	76.92
Sweden	84.03	84.08	84.42	84.76	85.09
United Kingdom	79.47	79.50	79.66	78.69	77.78
UE-27	78.02	78.01	79.38	79.35	79.58
UE-28	78.26	78.27	79.42	79.25	79.31

Source: Prepared by authors based on Eurostat.

**Table 4 healthcare-09-00353-t004:** Public Spending on Health (PSH) per capita (millions €—2010 constant prices).

Countries	2014	2015	2016	2017	2018	Av. Growth ^1^
Belgium	2682.34	2714.71	2709.27	2775.63	2772.66	0.83
Bulgaria	249.14	240.65	250.55	262.81	284.09	3.34
Czechia	972.01	965.52	971.32	1025.09	1123.60	3.69
Denmark	3844.35	3927.61	3987.43	4027.10	4099.46	1.62
Germany	3146.42	3210.81	3276.61	3385.52	3452.66	2.35
Estonia	601.44	638.84	667.84	701.62	740.97	5.35
Ireland	2734.87	2605.96	2687.05	2773.11	2927.68	1.72
Greece	767.19	784.11	851.57	837.77	810.39	1.38
Spain	1417.65	1501.04	1520.81	1541.44	1572.18	2.62
France	2776.21	2772.87	3039.87	3080.57	3107.18	2.86
Croatia	572.54	596.23	625.72	647.26	676.65	4.27
Italy	1700.34	1690.17	1689.25	1694.70	1711.66	0.17
Cyprus	605.86	603.63	623.43	643.78	692.48	3.40
Latvia	335.76	357.86	382.28	397.48	451.30	7.67
Lithuania	472.21	506.21	533.51	545.43	586.64	5.57
Luxembourg	3620.50	3592.53	3612.96	3647.63	3714.93	0.65
Hungary	510.20	524.34	547.03	558.47	580.51	3.28
Malta	1037.74	1074.11	1142.42	1196.00	1214.50	4.01
Netherlands	3305.55	3291.25	3312.30	3345.50	3392.76	0.65
Austria	2771.69	2770.82	2787.03	2843.80	2915.13	1.27
Poland	468.85	489.17	516.59	542.03	568.36	4.93
Portugal	969.44	987.16	985.18	1006.47	1053.27	2.10
Romania	279.26	281.45	300.65	335.67	385.77	8.41
Slovenia	1065.36	1100.15	1143.42	1147.90	1218.21	3.41
Slovakia	753.38	772.55	817.71	810.51	832.12	2.52
Finland	2624.41	2557.71	2525.71	2537.55	2557.66	−0.64
Sweden	3788.12	3865.65	3927.70	3970.17	4058.04	1.74
United Kingdom	2474.22	2498.51	2521.98	2510.31	2539.99	0.66
UE-27	1982.61	2010.66	2081.18	2124.96	2166.52	2.24
UE-28	2046.59	2074.00	2137.41	2174.63	2214.99	2.00

^1^ Average growth. Source: Prepared by authors based on Eurostat.

**Table 5 healthcare-09-00353-t005:** Public Expenditure on Health (PSH) (% GDP).

Countries	2014	2015	2016	2017	2018	Av. Growth ^1^
Belgium	7.92	7.90	7.83	7.92	7.81	−0.11
Bulgaria	4.51	4.16	4.14	4.16	4.34	−0.17
Czechia	6.28	5.93	5.83	5.86	6.25	−0.03
Denmark	8.56	8.61	8.53	8.44	8.45	−0.12
Germany	9.28	9.41	9.47	9.57	9.67	0.39
Estonia	4.61	4.79	4.86	4.85	4.92	0.31
Ireland	6.85	5.26	5.38	5.15	5.07	−1.78
Greece	4.56	4.65	5.05	4.90	4.66	0.10
Spain	6.38	6.50	6.40	6.31	6.31	−0.07
France	8.86	8.79	9.57	9.51	9.46	0.59
Croatia	5.55	5.61	5.64	5.58	5.62	0.07
Italy	6.69	6.59	6.49	6.40	6.39	−0.30
Cyprus	2.99	2.87	2.80	2.77	2.87	−0.12
Latvia	3.26	3.32	3.43	3.42	3.71	0.44
Lithuania	4.18	4.36	4.42	4.27	4.38	0.20
Luxembourg	4.55	4.42	4.36	4.42	4.45	−0.10
Hungary	4.74	4.68	4.77	4.65	4.58	−0.16
Malta	5.58	5.39	5.64	5.62	5.64	0.06
Netherlands	8.57	8.40	8.32	8.21	8.19	−0.38
Austria	7.67	7.67	7.66	7.68	7.71	0.04
Poland	4.49	4.49	4.60	4.60	4.58	0.09
Portugal	5.96	5.94	5.79	5.70	5.79	−0.17
Romania	3.97	3.86	3.92	4.05	4.43	0.47
Slovenia	6.05	6.12	6.16	5.91	6.02	−0.02
Slovakia	5.53	5.41	5.62	5.41	5.36	−0.17
Finland	7.63	7.42	7.15	6.98	6.95	−0.68
Sweden	9.20	9.08	9.15	9.14	9.27	0.07
United Kingdom	7.91	7.86	7.87	7.74	7.78	−0.13
UE-27	7.81	7.76	7.89	7.85	7.85	0.04
UE-28	7.83	7.78	7.88	7.83	7.84	0.01

^1^ Average growth. Source: Prepared by authors based on Eurostat.

**Table 6 healthcare-09-00353-t006:** Influence of the level of economic development on public expenditure on health (per capita) in the EU-28 Member States (2014–2018).

Regression Model with Panel Data				
Dependent Variable: Log(PSHpc ^1^)				
OLS Estimation				
Variable	Estimate 1		Estimate 2	
	Coefficient	t-Statistic	Coefficient	t-Statistic
Constant	−5.363574	−15.85622	−4.434975	−14.75819
Log(GDPpc ^2^)	1.253818	35.79316	-	-
Log(GDPpcn ^3^)	-	-	1.204515	37.14237
AR(1)	0.431266	7.885517	0.374002	7.597537
R-squared	0.923071		0.924544	
Adjusted R-squared	0.922017		0.923510	
Durbin-Watson stat	1.988998		1.980670	
F-statistic	875.9235		894.4489	
n	150		150	

^1^ PSHpc = Public spending on health per capita; ^2^ GDPpc: Gross Domestic Product per capita (level of economic development); ^3^ GDPpcn: Gross Domestic Product per capita net of the effect of income concentration within the EU-28 Member States. OLS = Ordinary least squares. Source: Prepared by authors based on Eurostat.

## Data Availability

Data from statistical sources cited in bibliographic references.

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
