# Peer review of "Dynamics of Public Spending on Health and Socio-Economic Development in the European Union: An Analysis from the Perspective of the Sustainable Development Goals"

_healthcare, 2021, doi:10.3390/healthcare9030353_

Round 1

Reviewer 1 Report

This is an interesting paper and the authors are quite correct that understanding health spending is a pressing issue. The analytical approach used here is good, and the paper has the potential to provide good insights into factors that drive spending in the EU (and other advanced economies). That said, there are two major issues that need to be addressed. The first is more straightforward - there are a lot of methods/variable descriptions presented in the Results section. These should be moved to the Materials and Methods section. For example, there is little description of the fixed effects model presented in Table 6 until you reach it - though this goes for all the results and their corresponding methods. I do not believe there is anything wrong with the content - just the way it is presented will need to be significantly restructured. This will help with the presentation of the article. The second issue is the lack of a literature review section. While the introduction does well to frame the problem, more theory, context, and interrogation of existing work is needed (i.e. work on health spending, variation in healthcare and its implications in Europe, economic development and sustainability/sustainable development are all potential topics that could be integrated here). This will help 'set the scene' of the research better. I think by addressing these two issues, the paper will be much stronger in presentation.

Author Response

Los autores desean agradecer al revisor los comentarios y sugerencias realizados sobre el documento titulado "Dinámica del gasto público en salud y desarrollo socioeconómico en la Unión Europea: un análisis desde la perspectiva de los Objetivos de Desarrollo Sostenible". Los autores los han tenido en cuenta en la segunda versión del artículo. Por supuesto, estas consideraciones nos han permitido mejorar sustancialmente el trabajo presentado.

Hemos ampliado la segunda sección de Materiales y métodos para explicar los métodos estadísticos y econométricos utilizados en el artículo (líneas 143 a 164). Asimismo, hemos enfocado adecuadamente el problema de investigación en la Introducción, manteniendo la estructura recomendada por la revista Healthcare a los autores para establecer los objetivos del trabajo (fin de la primera página, líneas 62 a 75, líneas 86 a 99).

Reviewer 2 Report

This study analyzed public spending of health and socio-economic development from the perspective of sustainable development goals.

However, in the introduction, the difference from the existing research, the Research Gap, should be more clearly indicated. To do this, it is necessary to supplement existing research-related literature studies.

In addition, existing studies required to establish an empirical analysis model need to be supplemented.

When the above contents are supplemented, conclusions on academic implications can be enriched.

Each table presented in the results properly uses data consistent with public spending on health socio-economic development in the EU.

Based on this, the discussion is developing research from the perspective of sustainable development goals.

Author Response

Los autores desean agradecer al revisor los comentarios y sugerencias realizados sobre el documento titulado "Dinámica del gasto público en salud y desarrollo socioeconómico en la Unión Europea: un análisis desde la perspectiva de los Objetivos de Desarrollo Sostenible". Los autores los han tenido en cuenta en la segunda versión del artículo. Por supuesto, estas consideraciones nos han permitido mejorar sustancialmente el trabajo presentado.

Hemos complementado la revisión de la literatura en la Introducción para enmarcar adecuadamente el estudio a realizar (final de la primera página, líneas 62 a 75, líneas 86 a 99) y ajustar con mayor precisión los objetivos planteados en el trabajo. Asimismo, hemos enriquecido las conclusiones sobre las implicaciones académicas del trabajo (líneas 578 a 588).